# Brassinolide Soaking Reduced Nitrite Content and Extended Color Change and Storage Time of *Toona sinensis* Bud during Low Temperature and Near Freezing-Point Temperature Storage

**DOI:** 10.3390/ijms232113110

**Published:** 2022-10-28

**Authors:** Xihang Xu, Chenchen Guo, Chunying Ma, Minghui Li, Ying Chen, Cunqi Liu, Jianzhou Chu, Xiaoqin Yao

**Affiliations:** 1School of Life Sciences, Hebei University, Baoding 071002, China; 2Institute of Life Sciences and Green Development, Hebei University, Baoding 071002, China; 3Key Laboratory of Microbial Diversity Research and Application of Hebei Province, Baoding 071002, China; 4College of Agronomy, Hebei Agricultural University, Baoding 071000, China

**Keywords:** brassinolide, near freezing-point temperature, low temperature, nitrite, *Toona sinensis*

## Abstract

Low temperatures are often used to preserve fruits and vegetables. However, low-temperature storage also causes problems, such as chilling injury, nitrite accumulation, and browning aggravation in plants. This study investigated the effects of brassinolide (BR,1.0 mg L^−1^) solution soaking, storage temperatures (−2 ± 0.5 °C, 4 ± 0.5 °C, and 20 ± 1 °C), and their combinations on nitrite content, color change, and quality of stored *Toona sinensis* bud. The results showed that low temperature (LT, 4 ± 0.5 °C) and near freezing-point temperature (NFPT, −2 ± 0.5 °C) storage effectively inhibited the decay of *T. sinensis* bud compared to room temperature (20 ± 1 °C, the control). The combined treatments of BR with LT or NFPT reduced nitrite content and maintained the color and the contents of vitamin C, carotenoids, saponins, β-sitosterol, polyphenol, anthocyanin, flavonoids, and alkaloids in *T. sinensis* bud. BR soaking delayed the occurrence of chilling injury during NFPT storage. Meanwhile, BR soaking enhanced the DPPH radical scavenging activity, ABTS activity, and FRAP content by increasing SOD and POD activity and the contents of proline, soluble, and glutathione, thus decreasing MDA and hydrogen peroxide content and the rate of superoxide radical production in *T. sinensis* bud during NFPT storage. This study provides a valuable strategy for postharvest *T. sinensis* bud in LT and NFPT storage. BR soaking extended the shelf life during LT storage and maintained a better appearance and nutritional quality during NFPT storage.

## 1. Introduction

*Toona sinensis* bud is a pollution-free vegetable with a unique flavor and rich nutrition, including high protein, carbohydrates, and many kinds of vitamins, minerals, flavonoids, polyphenols, alkaloids, and other important bioactive components [1,2]. Zhang et al. [3] reported that *T. sinensis* bud had a variety of diet therapy functions. In relevant experiments, the quercetin in *T. sinensis* bud can alleviate oxidative-stressful response in the body and the damage of hepatocyte organelles in diabetic mice, which is expected to become an auxiliary substance in the treatment of diabetes [4]. The aqueous extract of *T. sinensis* can induce the apoptosis of osteosarcoma cells and inhibit the abnormal growth of tumors [5]. However, due to high water content, vigorous physiological metabolism, and high respiration intensity, postharvest *T. sinensis* bud is prone to wilting, browning, defoliation, and decay [6]. Therefore, dealing with the fresh-keeping of postharvest *T. sinensis* bud is vital.

Low temperature (LT, 1–4 °C) and near freezing-point temperature (NFPT, −2–0 °C) are often used to preserve fruits and vegetables. However, LT or NFPT storage also causes problems, such as chilling injury, nitrite accumulation, and browning aggravation in horticultural crops [7,8,9]. The chilling damage on green bell pepper under low temperatures (3 °C) increased as the storage day increased (18-day) [10]. In the experiment performed by Mu et al. [11], with increased storage days (7-day), nitrite content in spinach stored at 4 °C increased first and then decreased, and nitrite content reached the maximum on the third day. The browning index of pear stored at 0–1 °C significantly enhanced with the increase of storage days [12]. However, the browning index of mung bean sprouts stored at 4 °C rose first and then reduced with the increase of storage day (7-day), reaching the maximum on the third day [13]. In addition, freezing treatment could destroy the organizational structure of fruits and vegetables, resulting in loss of juice during thawing and failure to maintain the original flavor of fruits and vegetables [14]. How to solve these problems of vegetables and fruits during LT or NFPT storage will be exciting research.

It has been reported that the application of exogenous hormones has a positive effect on plant resistance to stress [15]. Brassinosteroids (BRs), a safe, efficient, and non-toxic new plant hormone are found in many plants and exhibit distinctive biological effects on plant growth and development and resistance to biotic and abiotic stresses [16,17]. In recent years, BRs have also attracted extensive attention in storing postharvest horticultural products. Meanwhile, BRs have been proven to reduce the chilling injury symptoms of postharvest carambola and banana [18,19]. BRs have also been shown to alleviate the effects of abiotic stress on plants by modifying the ascorbate-glutathione cycle and secondary metabolites [16,20], modulation of stress markers [21], and modulation of antioxidants [22]. However, color change and nitrite accumulation during postharvest bud storage are critical problems. Whether BRs can minimize nitrite accumulation and maintain the color of postharvest buds during LT and NFPT storage, to improve the storage quality of postharvest buds and to better solve the problems of vegetables or fruits during LT or NFPT storage.

It is well known that nitrite has adverse effects on human health and is considered to be one of the essential carcinogenic factors in the human diet [23]. Therefore, the study aimed to explore the influences of brassinolide (BR) soaking, storage temperatures, and their combinations on nitrite content, color change, and quality of stored *T. sinensis* buds. We hypothesized that BR treatment could reduce nitrite accumulation, maintain the fresh color, and increase the quality of *T. sinensis* bud during LT or NFPT storage, which will help improve the shelf-life performance of vegetables or fruits stored under LT or NFPT conditions.

## 2. Results

### 2.1. BR Improves the Appearance Quality of Bud under Different Temperature Storage

The *T. sinensis* bud stored at 20 °C and 20 °C + BR gradually turned green and rotted after six days of storage, with the increase in storage days (Figure 1). There was no decay of buds during the whole storage at 4 °C and the combined treatments of 4 °C + BR. In addition, on days 6 and 9, the combined treatments of 4 °C + BR storage maintained a more original appearance color of buds, compared to the 4 °C storage alone. Buds stored at −2 °C showed visible chilling injury during the whole storage. Interestingly, the combined treatment of −2 °C + BR delayed the chilling injury time of buds, and the buds in −2 °C + BR storage maintained a normal appearance by days 3 and 6. By day 9, the chilling injury of buds was alleviated by −2 °C + BR storage, compared to the −2 °C storage alone.

### 2.2. Nitrite Content and Nitrate Reductase Activity in Buds

Figure 2 depicts the change in nitrite content and nitrate reductase activity in *T. sinensis* buds under different storage temperatures. The nitrite content in buds was reduced with the increase in storage time (Figure 2A). By days 3 and 6, the nitrite content in buds under 20 °C storage was lower than that in control (0 d), and the combined treatments of 20 °C + BR significantly enhanced the nitrite content, compared to the 20 °C storage, respectively. However, the combined treatments of 4 °C + BR and −2 °C + BR storage significantly decreased nitrite content in buds during the whole storage, compared to 4 °C and −2 °C storage, respectively. In addition, the combined treatments of 4 °C + BR and −2 °C + BR storage maintained a low value of nitrite content in buds compared to the control (day 0). By day 3, the nitrate reductase activity was significantly enhanced under the combined treatments of 20 °C + BR and −2 °C + BR and was significantly decreased under the combined treatments of 4 ℃ + BR storage, compared to the 20 °C, 4 °C, and −2 °C storage, respectively (Figure 2B). By day 6, the combined treatments of 4 °C + BR and −2 °C + BR storage significantly decreased the nitrate reductase activity, compared to the 4 °C and −2 °C storage, respectively.

### 2.3. Vitamin C, Carotenoid, Saponins, and β-Sitosterol Contents

As shown in Figure 3, the changes in vitamin C, carotenoid, saponins, and β-sitosterolin *T. sinensis* buds during different temperature storage were evaluated. By days 3 and 6, the vitamin C content in buds was greatly enhanced under the combined treatments of −2 °C + BR storage and was significantly decreased under the combined treatments of 4 °C + BR storage, compared to the 4 °C and −2 °C storage alone, respectively (Figure 3A). The combined treatments of −2 °C + BR markedly enhanced the vitamin C content in buds by day 9, compared to the −2 °C storage alone. By day 3, the combined treatments of 20 °C + BR storage significantly increased the carotenoid content in buds compared to the 20 °C storage, and the saponins content was markedly enhanced under the combined treatments of −2 °C + BR compared to the −2 °C storage (Figure 3B,C). The β-sitosterol content in buds was increased under different temperature storage, compared to the control (day 0), respectively (Figure 3D). By day 6, the β-sitosterol content in buds was significantly enhanced under the combined treatments of 20 °C + BR and −2 °C + BR, compared to the 20 °C and −2 °C storage alone, respectively.

### 2.4. Polyphenol, Polyphenol Oxidase, Anthocyanin, Flavonoids, and Alkaloid Contents

The combined treatments of −2 °C + BR significantly enhanced the polyphenol and anthocyanin contents in *T. sinensis* buds during the whole storage and decreased the polyphenol oxidase activity during the whole storage, compared to the −2 °C storage, respectively (Figure 4A–C). The flavonoids and alkaloid contents in buds were increased under the combined treatments of −2 °C + BR by day 6, compared to the −2 °C storage alone (Figure 4D,E). However, by day 9, there were no obvious changes in the flavonoids and alkaloid contents in buds under the combined treatments of 4 °C + BR and −2 °C + BR, compared to the 4 °C and −2 °C storage alone, respectively. The combined treatments of −2 °C + BR increased the PAL activity in buds by day 3 (Figure 4F). By day 6, the PAL activity in buds increased under different temperature storage compared to the control.

### 2.5. Antioxidant Enzyme Activity and Non-Enzymatic Antioxidant Contents

By day 6, the SOD and POD activity in *T. sinensis* buds were enhanced by the combined treatments of 20 °C + BR, compared to 20 °C storage (Figure 5). The combined treatments of −2 °C + BR enhanced the SOD activity, proline content, and soluble protein content in buds by day 3 and raised the POD activity, glutathione content, and soluble protein content by day 6, compared to the −2 °C storage, respectively (Figure 5 and Figure 6). By day 3, the combined treatments of 20 °C + BR, 4 °C + BR, and −2 °C + BR storage decreased the soluble sugar content in buds, compared to the 20 °C, 4 °C, and −2 °C storage, respectively (Figure 6C). In addition, the combined treatments of −2 °C + BR maintained similar SOD activity, glutathione content, soluble sugar, and soluble protein content in buds by day 9, compared to the control (day 0).

### 2.6. Antioxidant Capacity

The combined treatments of −2 °C + BR enhanced the ABTS activity and FRAP contents in *T. sinensis* by day 3 and increased the DPPH radical scavenging activity by day 6, compared to the −2 °C storage (Figure 7A–C). The DPPH radical scavenging activity and FRAP content in *T. sinensis* buds were decreased under the combined treatments of −2 °C + BR by day 9 and were enhanced under the combined treatments of 20 °C + BR storage by day 6, compared to the −2 °C and 20 °C storage, respectively. The combined treatments of 4 °C + BR raised the FRAP in buds by day 9 and enhanced the ABTS activity by day 6, compared to the 4 °C storage. The MDA content in buds decreased under the combined treatments of −2 °C + BR storage by days 6 and 9 and was decreased under the combined treatments of 20 °C + BR by days 3 and 6, compared to the −2 °C and 20 °C storage, respectively (Figure 7D). By days 3 and 6, the hydrogen peroxide level and the rate of O_2_^−^· production were both reduced by the combined treatments of −2 °C + BR in buds, compared to the −2 °C storage (Figure 7E). The rate of O_2_^−^· production in buds was decreased under the combined treatments of 4 °C + BR storage during the whole storage, compared to the 4 °C storage (Figure 7F). In addition, the combined treatments of 4 °C + BR and −2 °C + BR, 4 °C, and −2 °C storage maintained a higher level of FRAP content and ABTS activity in *T. sinensis* buds throughout the storage period compared to the control.

## 3. Discussions

Postharvest *T. sinensis* buds are highly susceptible to decay when stored at room temperature, which limits their storage time. Our results showed that cold storage at 4 °C extended the shelf life of buds. LT storage is one of the common methods to maintain the freshness of vegetables and fruits. NFPT storage has been employed to improve the storage time and quality of vegetables and fruits in recent years [24]. NFPT (−2.5 ± 0.2 °C) storage inhibited the chilling injury during apricot storage and maintained higher quality than conventional LT (10 and 5 °C) [25]. However, fluctuations in storage temperature during NFPT storage can lead to chilling injury of stored plants. The reason for chilling injury is the formation of ice crystals in plant tissue leading to cell dehydration, membrane leakage, and respiratory dysfunction [26]. 2,4-epibrassinolide improved the chilling tolerance of peach fruit by regulating PpCBF5-mediated membrane lipid metabolism and PpGATA12-mediated sucrose and energy metabolisms [27,28]. Chen et al. [29] reported that 2,4-epibrassinolide enhanced the chilling tolerance of loquat fruit by modulating cell walls and membrane fatty acid metabolism. The BR treatment delayed the chilling injury under NFPT storage of *T. sinensis* buds, indicating that BR treatment improved plant tolerance to cold stress during plant growth and post-harvest plant storage. Similar results showed that BR treatment alleviated the chilling injury of carambola [19] and banana [18] fruit under 4 °C and 2 °C storage, respectively.

Nitrate reductase is the key rate-limiting enzyme for reducing nitrate to nitrite in plants [8]. In the present results, the nitrite content was positively correlated with nitrate reductase activity (r = 3.83, *p* < 0.01) (Figure 8). Therefore, nitrate reductase activity has an important effect on nitrite formation during the storage of *T. sinensis* buds. Wei et al. [30] showed that hydrogen-rich water treatment reduced nitrate content in *Brassica campestris* seeding by increasing nitrate reductase activity. BR treatment reduced the rise of nitrite content in *T. sinensis* buds induced by LT and NFPT storage, and the nitrite content in buds stored at the combined treatments of 4 °C + BR and −2 °C + BR was lower than that of the control (day 0). This might be related to the decrease of nitrate reductase activity in *T. sinensis* buds during LT and NFPT storage by BR treatment.

The degradation of nutritional compounds during postharvest storage of buds has become a major limiting factor, which greatly limits its market value. Therefore, it is worth prolonging the shelf life and keeping/improving the postharvest storage quality of *T. sinensis* buds. Vitamin C is an important health-promoting compound in buds and is easily lost during storage [31]. In the present results, the vitamin C content in *T. sinensis* buds was enhanced by BR treatment during 4 °C storage and was higher than in the control. This finding is similar to the results obtained by Hu et al. [32], who reported that the content of vitamin C in *T. sinensis* buds decreased with storage time during 4 °C storage. In addition, vitamin C content in *T. sinensis* buds treated with BR decreased first and then increased during −2 °C storage. Cold storage positively influenced vitamin C in blond orange varieties [33]. The increased vitamin C content may result from cold stress resistance by day 9 during −2 °C storage. Carotenoids are the primary source of dietary vitamin A, which has antioxidant activity and helps prevent health problems such as cancer and cardiovascular/coronary heart disease [34,35]. Low temperature (7 °C and 12 °C) storage reduced the carotenoids accumulated in mango [36]. These results showed that the carotenoid content in *T. sinensis* was decreased by day 6 during 4 °C and −2 °C storage, compared to the control. Dietary saponins have been reported to prevent diseases caused by obesity [37]. The main food sources for humans to obtain sterols are bread, cereals, vegetables, etc., and sterols have mainly cholesterol-lowering properties [38]. The present results, showed no significant difference in vitamin C, carotenoid, saponins, and β-sitosterol contents between the combined treatments of 4 °C + BR and 4 °C storage alone by day 9. These findings revealed that BR treatment not only improved the appearance of *T. sinensis* buds when stored at 4 °C but also did not alter the contents of vitamin C, carotenoid, saponins, and β-sitosterol. The same results were found during −2 °C storage.

Vegetables and fruits are rich in polyphenols, and their content is influenced by storage. Materska et al. [39] found that the total phenolic content of lettuce was reduced by 55% at 4 °C storage for 7 days. The present results were similar to those of Materska et al. in that the polyphenol content in *T. sinensis* buds was decreased by day 3 during different temperature storage, compared to the control. BR increased the polyphenol content in *T. sinensis* buds during storage. The quality of apple juice was changed during storage in association with polyphenol oxidase activity [40]. Polyphenols serve as substrates for lychee polyphenol oxidase, and polyphenol oxidase degrades anthocyanins in the presence of polyphenols [41]. Anthocyanins are the primary color-presenting molecules in plants, and they serve a key role in preventing coronary heart disease, cancer and anti-inflammatory actions [34]. The anthocyanin content in *T. sinensis* buds was increased by BR treatment during storage. The results were similar to that BR treatment enhanced the anthocyanin content in small black bean sprouts by 0.56-fold after 5 days of 4 ± 1 °C storage, compared to the control [42]. The increase in polyphenol and anthocyanin contents in *T. sinensis* buds may be due to decreased polyphenol oxidase activity by BR treatment. Moreover, anthocyanins are important flavonoids in plants, which help protect plants from the damaging effects of reactive oxygen species by serving as free radical scavengers under salt stress [43]. Our study suggested that BR treatment increased the flavonoid content in *T. sinensis* buds under 4 °C and −2 °C storage, suggesting that BR treatment may reduce the cold damage during LT and NFPT storage by increasing the flavonoid content. The *T. sinensis* leaves are rich in alkaloids, which are important nutrients [44]. PAL is a crucial enzyme in producing secondary metabolites, such as flavonoids, alkaloids, and phenols [42]. Flavonoid content and PAL activity were found to have a positive relationship. (r = 4.36, *p* < 0.01) (Figure 8). The alkaloid content in *T. sinensis* buds was increased by BR under −2 °C storage, which may be related to the increased PAL activity. Other studies also found that BR treatment improved PAL activity in tomatoes during 1 °C storage [45].

Exogenous BR treatment can improve the tolerance of fruits and vegetables to cold stress, such as peaches [46], grapevines [47], green bell pepper [10], etc. BR treatment mainly improves plant tolerance to cold stress by increasing plant antioxidant enzyme activity and non-enzymatic antioxidant contents, reducing reactive oxygen species and membrane lipid peroxidation. We observed that the BR treatment delayed the appearance of chilling injury under −2 °C storage and maintained a more pristine appearance by day 9 during 4 °C storage than 4 °C storage (Figure 1). Subsequently, the effects of BR treatment on the antioxidant capacity were studied by measuring the antioxidant enzyme activity and non-enzyme antioxidant contents of *T. sinensis* buds at different storage temperatures. In the present results, the SOD activity in *T. sinensis* buds was increased under BR treatment by day 9 during 4 °C storage. The results were similar to those of carambola fruit treated with BR during 4 °C storage, which increased SOD activity to delay the chilling harm [19]. BR treatment increased the SOD activity, proline content, and soluble protein content by day 3 during −2 °C storage and increased the POD activity, glutathione content, and soluble protein content by day 6 during −2 °C storage. These findings showed that BR delayed the chilling injury in −2 °C storage by enhancing the antioxidant enzyme activity and non-enzymatic antioxidant contents. Wang et al. [10] observed that BR treatment induced the activity of antioxidant enzymes such as catalase, peroxidase, ascorbate peroxidase, and glutathione reductase in green bell pepper during 3 ℃ storage, which may alleviate the chilling injury and improve the storage quality. Proline is a multifunctional amino acid that aids in osmotic control and cell membrane stability between the cytoplasm and osmotic vesicles [46]. BR treatment increased the proline content and decreased the MDA content of tomatoes during 1 ± 0.5 °C storage, which improved the chilling stress tolerance of tomato fruit [48]. The present results were similar in that the MDA content in *T. sinensis* buds was decreased by BR treatment during −2 °C storage. In addition, the increased GSH content after BR treatment inhibited reactive oxygen species contents and lipid peroxidation [47]. Soluble sugar content was related to the cold tolerance of bananas [49]. However, we found that the soluble sugar content in *T. sinensis* buds was decreased by BR treatment during 4 °C and −2 °C storage on day 3, and BR treatment maintained a similar soluble sugar content to the control during 4 °C and −2 °C storage by day 3. This might be related to the fact that BR treatment alleviated the response of *T. sinensis* buds to cold stress, thus reducing sugar metabolism under LT and NFPT storage.

Finally, we evaluated the antioxidant activity of *T. sinensis* buds by FRAP, ABTS, and DPPH radical scavenging activity methods. We measured the changes in MDA content, hydrogen peroxide content, and the rate of O_2_^−^· production of *T. sinensis* buds under different storage temperatures after BR treatment. There was a negative correlation between the content of hydrogen peroxide and the DPPH radical scavenging activity (r = −0.370, *p* < 0.01) and ABTS activity (r = −0.486, *p* < 0.01) (Figure 8). These findings suggested that BR treatment improved the antioxidant capacity of *T. sinensis* buds under LT and NFPT storage, thus reducing the reactive oxygen species contents. The increased antioxidant capacity of BR treatment might be the main reason BR treatment improved the shelf life of *T. sinensis* buds under 4 ℃ storage and delayed the emergence of chilling injury under −2 °C storage. The results agreed with a previous study that BR treatment effectively inhibited MDA and reactive oxygen species generation and alleviated chilling injury during 4 °C storage and cold chain circulation of carambola fruit [19]. Other studies found similar results that the accumulation of MDA and hydrogen peroxide was significantly reduced by gibberellic acid treatment in *T. sinensis* buds during 4 ± 1 °C storage and effectively enhanced the tolerance of buds to chilling stress [32].

In summary, BR treatment delayed the occurrence of chilling injury in *T. sinensis* buds during −2 °C storage. In addition, the BR treatment maintained a more original appearance color and decreased the nitrite content in buds during 4 °C and −2 °C storage. By day 9, the nutritional qualities of buds such as vitamin C, carotenoids, saponins, β-sitosterol, polyphenol, anthocyanin, flavonoids, and alkaloid contents were not significantly different between the combined treatments of 4 °C + BR storage and 4 °C storage. By day 6, the combined treatments of −2 °C + BR enhanced the β-sitosterol and anthocyanin contents compared to the combined treatments of 4 °C + BR storage but did not differ substantially in vitamin C, carotenoids, saponins, and polyphenols contents. Furthermore, BR treatment improved antioxidant capacity by increasing antioxidant enzyme activity and non-enzymatic antioxidant contents, which reduced the occurrence of chilling injury of *T. sinensis* under −2 °C storage. This study confirmed our hypothesis that BR treatment reduced nitrite accumulation, extended color change, and increased the quality of *T. sinensis* buds during LT and NFPT storage. It provides a convenient and economical method for vegetable and fruit storage.

## 4. Materials and Methods

### 4.1. Plant Materials and Treatments

*T. sinensis* buds were harvested from Zhuozhou *T. sinensis* growing base, Hebei, China, in April 2021. Brassinolide was provided by ANPEL Laboratory Technologies (Shanghai, China) Inc. *T. sinensis* buds with uniform size, color and absence of disease, and other defects were chosen and conveyed to the laboratory instantly after harvesting. Buds were disinfected for 4 min in 1% sodium hypochlorite, washed, and aired. Half of the buds were soaked in BR solution with 1.0 mg L^−1^ for 2 h, while the other half was immersed in distilled water as the control group. The concentration and soaking time of BR was based on our preliminary experiments. The NFPT of *T. sinensis* buds was determined to be −2 ± 0.5 °C, according to Fan et al. [50] and Yang et al. [51]. After soaking, the storage temperatures were designed with 20 ± 1 °C as room temperature (the control), 4 ± 0.5 °C as LT, and −2 ± 0.5 °C as NFPT for 9 days, respectively. All samples were taken every three days. Some samples were frozen at −80 °C to determine the enzyme activity, and others were kept in an electric blast drying oven at 105 °C for 15 min, then dried to constant weight at 56 °C and ground using mortar. Each treatment group was repeated four times.

### 4.2. Nitrite and Nitrate Reductase Activity

Nitrite content was assayed by the National Food Safety Standard of the People’s Republic of China (GB5009.33-2010). The bud samples (0.2 g) were ground in a saturated borax solution (5 mL) and cultivated for 15 min in a boiling water bath. The mixture contained potassium ferrocyanide, zinc acetate, and sample extraction and was centrifuged at 4507× *g* for 10 min. The supernatant was measured at 538 nm, and the result was shown as μg g^−1^ fresh weight (FW).

Nitrate reductase activity was measured based on the previous report [8]. The fresh buds (0.2 g) were ground in enzyme extract (5 mL) containing phosphate buffer (25 mmol L^−1^, pH 7.4, 0.1 mol L^−1^), EDTA (1 mmol L^−1^), and cysteine (10 mmol L^−1^). After centrifugation, the mixture, including 0.2 mL supernatant, 0.5 mL potassium nitrate, and 0.3 mL NADH, was cultured for 30 min at 25 °C. 2 mL of sulfonamide reagent and 2 mL of *α*-naphthylamine were added to the mixture and were shook well. After 15 min, the absorbance value was measured at 520 nm, and the result was shown as U g^−1^ h^−1^ FW.

### 4.3. Vitamin C, Carotenoid, Saponin, and β-Sitosterol

Vitamin C was determined according to the 24-dinitrophenylhydrazine method [31]. Dried samples (0.2 g) were extracted with oxalic acid (1%, 8 mL). An appropriate amount of activated carbon was mixed with 10 mL diluted extract to vibrate. After filtering, the supernatant (5 mL) was mixed with 24-dinitrophenylhydrazine (2%, 1 mL) and thiourea solution (2%, 5 mL) and was cultivated for 3 h at 37 ℃. H_2_SO_4_ (85%) was slowly added to the reaction solution and was instantly shaken. Vitamin C was determined by measuring the absorbance at 500 nm and was indicated as mg g^−1^ dried weight (DW).

Carotenoid content was determined based on the method described by Ghassemi-Golezani et al. [43]. The sample was soaked in acetone (80%) until colorless. The result was indicated as μg g^−1^ FW.

Saponin was determined based on the previous report [52]. Dried samples were mixed with methanol solution (5 mL) in an ultrasonic bath for 30 min. The solvent was volatilized from the extract (0.2 mL), and vanillic aldehyde-glacial acetic acid and perchloric acid were added to the residue. After shaking, the reaction was carried out at 80 °C for 30 min and was terminated on an ice bath for 10 min. The solution absorbance value was measured at 540 nm, and the result was indicated as mg g^−1^ DW.

β-sitosterol content was measured based on the previous report with minor modifications [53]. Dried samples were immersed in an ethyl acetate solution for 12 h. After centrifuging, the supernatant (4 mL) was evaporated in a water bath. The residue was dissolved in 5 mL sulfuric acid at 50 °C for 4 min. The solution absorbance value was measured at 407 nm, showing the result as mg g^−1^ DW.

### 4.4. Polyphenol, Polyphenol Oxidase (PPO), and Anthocyanin

Polyphenol content was determined using the method described by Jiang et al. [54] with a small amount of adjusting. Dried samples (0.2 g) were extracted with 10 mL boiling water for 15 min, then were centrifuged at 2535× *g* for 10 min. The reaction solution included 1 mL of supernatant, 3 mL phosphoric acid buffer (pH 6.8), and 1 mL of 0.1 mol L^−1^ iron potassium tartrate. The absorbance was taken at 540 nm after blending, and the data was calculated using a linear equation of the catechin standard curve. Polyphenol content was expressed as μg g^−1^ DW.

PPO activity was measured based on the previous report [40]. Fresh samples (0.2 g) were homogenized with 5 mL of pre-cooled phosphoric acid buffer (pH 6.5) and were centrifuged for 10 min at 2535× *g*. The enzyme extract (0.1 mL) was added to the reaction mixture, including 0.1 mol L^−1^ phosphoric acid buffer with pH 6.5 and 0.01 mol L^−1^ catechol solution, followed by a water bath for 30 min at 30 ℃. The reaction was terminated with 20% trichloroacetic acid. The absorbance value was measured at 410 nm, and the result was shown as U g^−1^ h^−1^ FW.

Anthocyanin content was analyzed with 5 mL of 1% HCl-methanol solution in darkness for 3 days [32]. The absorbance value was measured at 532 and 600 nm, and the result was shown as U g^−1^ h^−1^ FW.

### 4.5. Flavonoids, Alkaloid, and Phenylalanine Ammonia Lyase (PAL) Activity

Flavone content was evaluated according to AlCl_3_ colorimetry [32]. About 0.2 g sample was mixed with an alcohol solution (5 mL, 50%) for 30 min in an ultrasonic bath. The reaction mixture contained supernatant (2 mL), acetic acid-sodium acetate buffer (2 mL, pH 5.4), and AlCl_3_ (4 mL, 1.5%). The absorbance was measured at 415 nm after 30 min, and the flavone content was calculated as g 100 g^−1^ DW.

The total alkaloid was assayed according to the report with tiny modifications [53]. Dried samples (0.2 g) were extracted with ammonium hydroxide (0.5 mL, 25%) and chloroform (5 mL) at 37 °C for one hour. The 2 mL extract was mixed with citric acid-sodium citrate (5 mL, pH 5.4), bromothymol blue (0.5 mL, 0.1%), and chloroform (3 mL). After one hour, the absorbance was measured at 416 nm, and the total alkaloid was shown as mg g^−1^ DW.

PAL activity was measured based on the report [42]. The fresh samples (0.2 g) were extracted with 0.05 mol L^−1^ boric acid buffer (pH 8.7) containing mercaptoethanol (5 mmol L^−1^) and polyvinylpolypyrrolidone (0.1%). The supernatant (0.5 mL) was mixed with the 1 mL phenylalanine (0.02 mol L^−1^) and 3 mL borate buffer and was cultured at 30 °C for 0.5 h. The absorbance value was measured at 290 nm, and the result was indicated as U kg^−1^ h^−1^ FW.

### 4.6. Superoxide Dismutase (SOD), Peroxidase (POD), Proline

SOD and POD were determined using the NBT method and the guaiacol method, respectively [32]. Fresh sample (0.5 g) was extracted in 10 mL sodium phosphate buffer (0.1 mol L^−1^) containing polyvinylpolypyrrolidone (2%, *w*/*v*). The extracted solution pH for SOD and POD were 6.8 and 6.4, respectively. The homogenate was centrifuged at 12,000× *g* for 30 min at 4 °C. SOD reaction solution included of 0.05 mol L^−1^ sodium phosphate buffer (1.7 mL), 0.014 mol L^−1^ methionine (0.3 mL), 0.75 mmol L^−1^ NBT (0.3 mL), 1.0 μmol L^−1^ EDTA (0.3 mL), 20.0 μmol L^−1^ riboflavin (0.3 mL), and enzyme extract (0.1 mL). SOD activity was analyzed by monitoring 50% inhibition of nitro blue tetrazolium (NBT) photochemical reduction. The solution was measured at 560 nm, and the result was shown as U g^−1^ FW min^−1^.

POD reaction solution included 0.05 mol L^−1^ sodium phosphate buffer (2.7 mL), of 0.02 mol L^−1^ H_2_O_2_ (0.1 mL), 0.02 mol L^−1^ guaiacol (0.1 mL) as a substrate, and 0.1 mL of enzyme extract. The solution was measured at 470 nm, and the result was shown as U kg^−1^ FW min^−1^.

Proline was determined based on the method described by Xi et al. [47]. The dried sample (0.2 g) was immersed in sulfosalicylic acid. After centrifugation, supernatant (0.5 mL) was mixed with glacial acetic acid (0.5 mL) and ninhydrin reagent (0.5 mL) and was incubated for 60 min at 100 °C. After cooling, toluene (1 mL) was added to the solution, and the absorbance value was recorded at 520 nm.

### 4.7. Glutathione, Soluble Sugar, and Soluble Protein

Glutathione was determined based on the method described by Ahmad et al. [20]. The sample was extracted with phosphoric acid buffer and DTNB solution. After 10 min, the supernatant absorbance value was determined at 412 nm and was indicated as mmol g^−1^ FW.

Soluble sugar was measured based on the report [55]. Soluble sugar was extracted with distilled water (5 mL) and cultured in a boiling bath for 20 min. The reaction solution contained extract (0.1 mL), ethyl-alcohol (80%, 0.9 mL), distilled water (1 mL), and anthrone reagent (5 mL). The value was measured at 625 nm wavelength using a spectrophotometer, and the result was presented in terms of mg g^−1^ DW.

Soluble protein was measured with Coomassie bright blue [56]. The sample was immersed in phosphate buffer (pH 7.0, 0.1 mol L^−1^, 5 mL) and was extracted in an ultrasonic bath. After centrifugation, the supernatant was added to 5 mL Coomassie bright blue solution. The absorbance value was measured at 595 nm, and the result was presented as mg g^−1^ DW.

### 4.8. 2,2-Diphenyl-1-Picrylhydrazyl (DPPH) Radical Scavenging, Ferric Reducing Ability of Plasma (FRAP), and 2,2′-Azinobis-(3-Ethylbenzothiazoline-6-Sulfonic Acid (ABTS)

DPPH radical scavenging activity was performed, as reported by Xue et al. [42]. The sample (0.1 g) was extracted with anhydrous ethanol. After centrifugation. The extract solution (2 mL) was added into DPPH solution (0.04 mg/mL, 2 mL) for 30 min in the dark. The absorbance value was measured at 517 nm. The calculation formula is as follows:Scavenging activity (%) = (A0 − A1 + A2)/A0 × 100%

A0 is the absorption value of the control (the ethanol replaced the sample solution); A1 is the absorption value of the sample; A2 is the absorption value of the sample under similar conditions as A1 with the ethanol replaced the DPPH solution.

FRAP and ABTS radical cation decolorization were determined using the method described by Deng et al. [31]. FRAP content was determined by adding 0.1 mL of sample solution into 0.3 mL of distilled water and 3 mL of FRAP (37 °C) solution, and the solution was cultured for 30 min in the dark. The absorbance value was measured at 592 nm and was expressed as mg g^−1^ FW. ABTS activity was measured by mixing a 20 μL sample solution with 4 mL ABTS mixture, and the solution was cultured for 6 min at room temperature in the dark. The absorbance was measured at 734 nm, and the results were expressed as mg g^−1^ FW.

### 4.9. Malondialdehyde (MDA), Hydrogen Peroxide, the Rate of Superoxide Radical (O_2_^−^·) Production

MDA was measured by the thiobarbituric acid (TBA) reaction [57]. Samples were extracted with trichloroacetic acid (TCA, 5 mL 20%). The reaction solution included the supernatant (2 mL), TCA (2 mL), and butylated hydroxytoluene (4%, 1 mL). The solution was measured at 532, 450, and 600 nm, and the result was shown as μmol g^−1^ FW.

Hydrogen peroxide was measured based on the report [57]. Fresh samples (0.2 g) were extracted with acetone (5 mL) at 4 °C. The reaction solution included the extract (0.1 mL), titanium sulfate solution (1 mL, 5%), and concentrated ammonia water (3 mL). After centrifuging, the residue was dissolved with 2 mol L^−1^ sulfuric acid solution. The supernatant was measured at 415 nm, and the result was shown as μg g^−1^ FW.

The rate of O_2_^−^· production was measured as described by Xi et al. [47]. Fresh sample (0.5 g) was extracted with potassium phosphate (pH 7.8, 5 mL). The reaction solution included supernatant (0.5 mL), hydroxylamine hydrochloride (0.5 mL, 10 mM), phosphate buffer (pH 7.8, 1 mL, 65 mM). After 20 min, sulfanilamide (8.5 mM) and a-naphthylamine (3.5 mM) were added to the mixture. After 20 min, the absorbance was read at 530, and the result was shown as mol^−1^ g^−1^ FW min^−1^.

### 4.10. Statistical Analysis

Random sampling was used in all experiments. All data were obtained from the average of four repeated experiments. All data were statistically determined using SPSS 21.0, and the findings were analyzed using one-way ANOVA and Duncan’s test to see if there were significant differences (*p* < 0.05). The Pearson correlation coefficient examined the relationship between anthocyanin, nitrate, nitrite reductase, and quality indices. The correlation data was made into a heat map by Origin 2019b software.

## Figures and Tables

**Figure 1 ijms-23-13110-f001:**
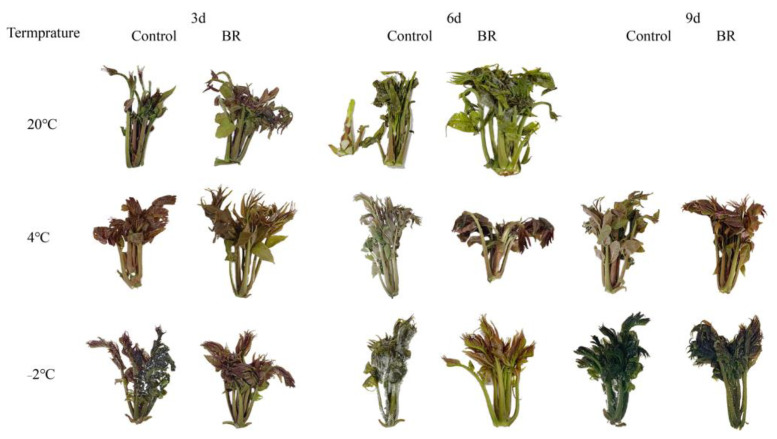
Appearance changes of *T. sinensis* as affected by BR treatment during different temperature storage.

**Figure 2 ijms-23-13110-f002:**
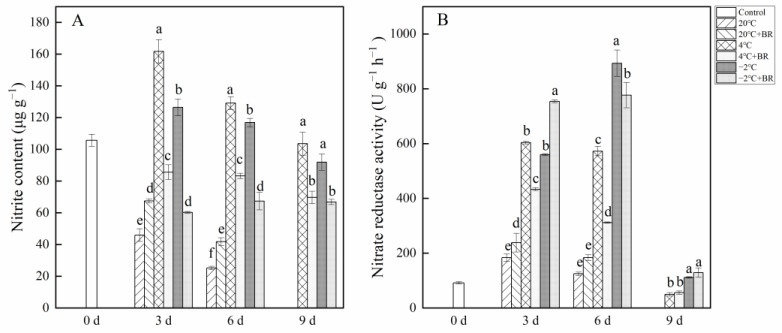
Effects of BR treatment on nitrite content (**A**) and nitrate reductase activity (**B**) of *T. sinensis* during different temperature storage. Bars with different letters are significantly different at the same storage time under different temperatures and BR treatment (*p* < 0.05). Values are the mean of four replicates ± SE.

**Figure 3 ijms-23-13110-f003:**
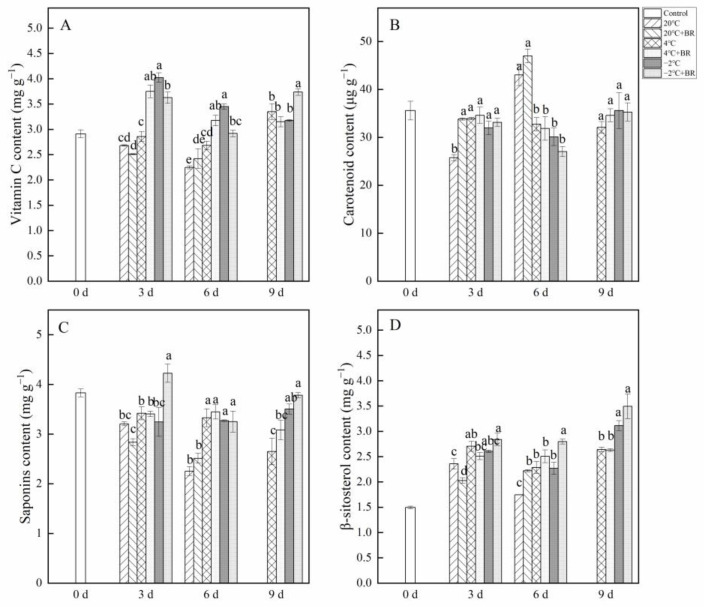
Effects of BR treatment on vitamin C (**A**), carotenoid (**B**), saponins (**C**), and β-sitosterol (**D**) content of *T. sinensis* during different temperature storage. Bars with different letters are significantly different at the same storage time under different temperatures and BR treatment (*p* < 0.05). Values are the mean of four replicates ± SE.

**Figure 4 ijms-23-13110-f004:**
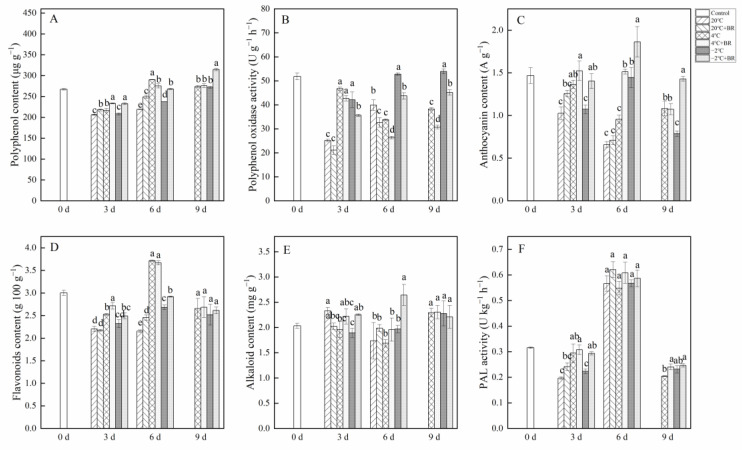
Effects of BR treatment on polyphenol content (**A**), polyphenol oxidase activity (**B**), anthocyanin content (**C**), flavonoids content (**D**), alkaloid content (**E**), and PAL activity (**F**) of *T. sinensis* during different temperature storage. Bars with different letters are significantly different at the same storage time under different temperatures and BR treatment (*p* < 0.05). Values are the mean of four replicates ± SE.

**Figure 5 ijms-23-13110-f005:**
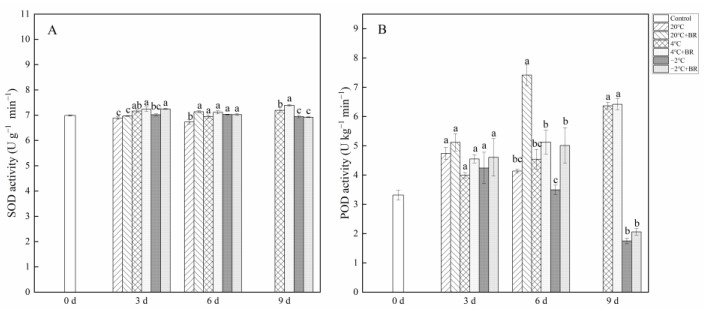
Effects of BR treatment on SOD activity (**A**), POD activity (**B**) of *T. sinensis* during different temperature storage. Bars with different letters are significantly different at the same storage time under different temperatures and BR treatment (*p* < 0.05). Values are the mean of four replicates ± SE.

**Figure 6 ijms-23-13110-f006:**
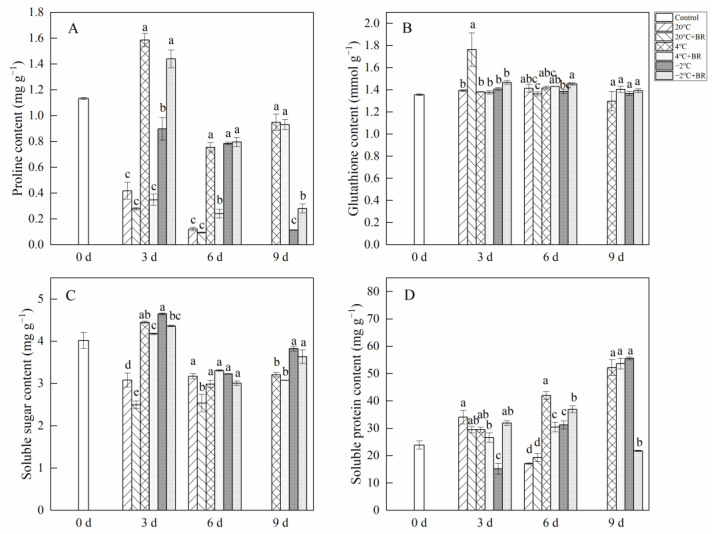
Effects of BR treatment on proline content (**A**), glutathione content (**B**), soluble sugar content (**C**), and soluble protein content (**D**) of *T. sinensis* during different temperature storage. Bars with different letters are significantly different at the same storage time under different temperatures and BR treatment (*p* < 0.05). Values are the mean of four replicates ± SE.

**Figure 7 ijms-23-13110-f007:**
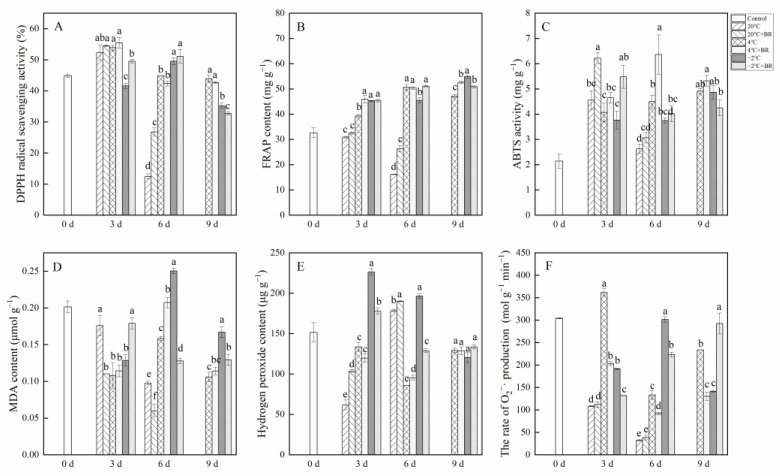
Effects of BR treatment on DPPH radical scavenging activity (**A**), FRAP (**B**), ABTS (**C**), MDA content (**D**), hydrogen peroxide content (**E**), and the rate of superoxide radical (O_2_^−^·) production (**F**) of *T. sinensis* during different temperature storage. Bars with different letters are significantly different at the same storage time under different temperatures and BR treatment (*p* < 0.05). Values are the mean of four replicates ± SE.

**Figure 8 ijms-23-13110-f008:**
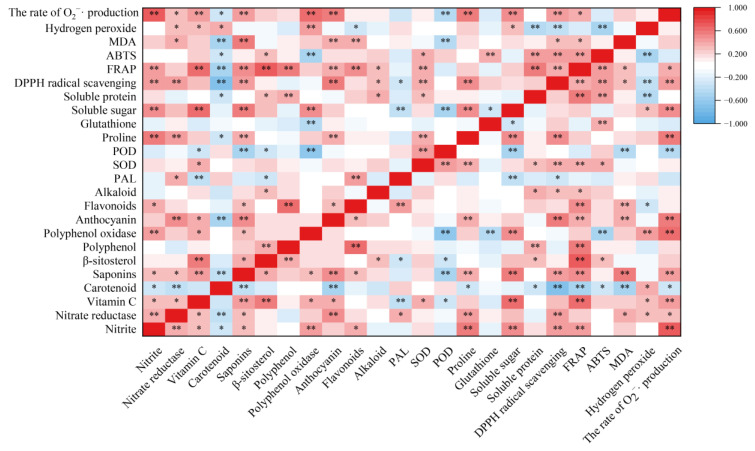
Pearson’s correlation coefficient between the nutrition quality and antioxidant capacity of *T. sinensis* during different temperature storage after BR treatment. * and ** indicated the significant correlations at the 0.05 and 0.01 levels, respectively.

## Data Availability

Not applicable.

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
