# Peer review of "Brassinolide Soaking Reduced Nitrite Content and Extended Color Change and Storage Time of *Toona sinensis* Bud during Low Temperature and Near Freezing-Point Temperature Storage"

_ijms, 2022, doi:10.3390/ijms232113110_

Round 1

Reviewer 1 Report

The comments are attached.

Author Response

Specific performance in the following:

  1. The introduction is superficial and lacks in depth. A description or review of current study of T. sinensis bud storage will be very helpful to the significant paper. In particular, discussing what are the situation and challenges for now.

Yes, we added the discussing what are the situation and challenges for now.

“However, due to high water content, vigorous physiological metabolism, and high respiration intensity, postharvest T. sinensis bud is prone to wilting, browning, defoliation, and decay [5]. Therefore, it is vital to deal with the fresh-keeping of postharvest T. sinensis bud.” See line 41-44

“How to solve these problems of vegetables and fruits during LT or NFPT storage will be exciting research.” See line 57-59

“Whether BRs can minimize nitrite accumulation and maintain color of postharvest buds during LT and NFPT storage, to improve the storage quality of postharvest buds and to better solve the problems of vegetables or fruits during LT or NFPT storage.” See line 70-73.

  1. It will be good if the authors performed the gene expression or metabolism analysis of T.sinensis bud under the BR soaking and storage temperatures. That helps to understand the results obtained from this study.

Sorry, we did not measure gene expression and metabolism analysis. The proposal is very good and gives us a good idea. Next, we will further study the gene expression and metabolism analysis of T. sinensis bud under the BR soaking and storage temperatures. Thanks again.

But, brassinosteroids are steroidal plant hormones, so we submitted the paper to the "Role of Steroids and Triterpenoids in Plant Growth, Development and Stress Response" special issue. We hope that the paper is published.

  1. More detail is required in the “Materials and Methods” section to enable anyone to be able to undertake this study in the way that it was originally undertaken, such as the determination of nitrate reductase activity, phenylalanine ammonia lyase (PAL) activity, etc. Especially, how to define and calculate enzyme activity?

Yes, we added the detail in the “Materials and Methods” section.

  1. As mentioned by authors say, “In the present results, the nitrite content was positively correlated with nitrate reductase activity (r = 3.83, P < 0.01) (Fig. 3).” on the page 7. In general, the nitrate contents in plants were decreased with the increase of nitrate reductase activity. Please explain the phenomenon in this study.

Yes, nitrate reductase is the key rate-limiting enzyme for reducing nitrate to nitrite in plants. In the study, according to Pearson correlation coefficient analysis, nitrate reductase activity and nitrite content were positively correlated.

So, we concluded that the decrease in nitrite contents (not nitrate contents) might be related to the decrease of nitrate reductase activity in T. sinensis buds during LT and NFPT storage by BR treatment. In general, the nitrite contents (not nitrate contents) in plants were decreased with the increase of nitrate reductase activity.

  1. In the Figure 1, the color change of T.sinensis bud with different treatment were not obvious as authors described.

On days 6 and 9, T. sinensis bud’s color changes from red to green under normal temperature storage, while the original red color of T. sinensis buds remains at low-temperature storage.

  1. Referencing has to be improved. The authors reference on general knowledge, the literature survey has to be strengthened and go into detail.

Yes, we changed the referencing in revised paper.

  1. The current manuscript have many grammatical errors, therefore the paper could benefit from proof reading/ language editing. Such as “content” should be “contents” on the page 9, ……

Yes, we revised it and checked the grammatical errors in revised paper.

Reviewer 2 Report

Dear Authors,

I reviewed your article titled in (Brassinolide soaking reduced nitrite content and extended color change and storage time of Toona sinensis bud during low temperature and near freezing-point temperature storage). Overall, the data presented here is valuable to those working in this field and demonstrates the effectiveness of a relatively simple intervention that could be applied on a wider scale, especially in the field of postharvest technology. The article is written and organized well. A thorough revision of the English grammar, and sentence structure, and simple editing of some parts need to be undertaken. There are some other major points that should be addressed in the individual sections, which I have specified in the attached file. 

Best wishes

Author Response

Thank you very much for your comments with regard to our manuscript “Brassinolide soaking reduced nitrite content and extended color change and storage time of Toona sinensis bud during low temperature and near freezing-point temperature storage (ijms-1981806)”. We have revised our manuscript according to receiving comments and responded to all comments raised by the reviewers and editor in the online review forum. Meanwhile, we submitted a revised version of our manuscript. The changes in the revised paper were marked with red color text.

Our responses to reviewers’ comments are in the uploaded PDF file.

Round 2

Reviewer 1 Report

The author has revised the manuscipt according to comments, now it can be accepted as far as I am concerned.

Reviewer 2 Report

Dear Authors,

Thank you so much for your responses. Much of it has been addressed.

The authors made a great effort, and the MS is now suitable for publication.